

# Inorganic and organic geochemical fingerprinting of sediment sources and ocean circulation on a complex continental margin (São Paulo Bight, Brazil)

Michel Michaelovitch de Mahiques[1], Till Jens Jörg Hanebuth[2], Renata Hanae Nagai[3], Marcia Caruso Bícego[1], Rubens Cesar Lopes Figueira[1], Silvia Helena de Mello e Sousa[1], Leticia Burone[4], Paula Franco-Fraguas[4], Satie Taniguchi[1], Alexandre Barbosa Salaroli[1], Gilberto Pereira Dias[1], Denise Menezes Prates[1], Maria Eugenia Fernandes Freitas[1]

[1]Instituto Oceanográfico, Universidade de São Paulo, 05508-120 Praça do Oceanográfico, 191 São Paulo, SP, Brasil

[2]School of Coastal and Marine Systems Sciences, Coastal Carolina University, 290 Allied Drive, Conway, SC 29526, USA

[3]Centro de Estudos do Mar, Universidade Federal do Paraná, 83255-976 Av. Beira-mar, s/n, Pontal do Paraná, Brasil

[4]Facultad de Ciencias, Universidad de La Republica, Iguá, 4225, Montevideo, 11400, Uruguay

*Correspondence to*: Michel M. de Mahiques (mahiques@usp.br)

**Abstract.** In this study, we use inorganic (metals) and organic (bulk and molecular) markers in sediment samples of the southeastern Brazilian margin to investigate the response of geochemical fingerprints to the complex hydrodynamic processes present in the area. Results indicate the potential of export of terrigenous siliciclastic and organic constituents to
the upper slope, even in an area with limited fluvial supply.

Metal contents and especially the $\ln(Ti/Al)$ and $\ln(Fe/K)$ ratios make it possible to recognize the extension of shelf sediments toward the upper slope. Potassium, here expressed as $\ln(K/Sc)$ and $\ln(K/Al)$ ratios used as proxies of illite–kaolinite variations, proved to be an important parameter, especially because it allowed us to decipher the imprint of the northward flow of the Intermediate Western Boundary Current (IWBC) in comparison to the southward flows of the Brazil Current
(BC) and Deep Western Boundary Current (DWBC).

Using organic matter analyses, we were able to evaluate the extent of terrestrial contributions to the outer shelf and slope, even without the presence of significant fluvial input. In addition, molecular markers signify a slight increase in the input of $C_4$-derived plants to the slope sediments, transported from distant areas by the main alongshore boundary currents, indicating



that the terrestrial fraction of the organic matter deposited on the slope has a distinct origin when compared to shelf sediments.

# 1 Introduction

Sedimentary processes at continental margins are controlled by a complex interplay of factors that act on local, regional and global scales (Pratson et al., 2007). The physical and chemical processes that govern a depositional environment leave particular imprints in the sediments. Thus, the spatial distribution of a characteristic sediment composition can be used to decipher the environmental history of an area, in particular in terms of material supply and oceanographic conditions (Boggs, 2005; Nittrouer et al., 2007; Stow et al., 2008; Hernandez-Molina et al., 2009).

Sedimentation along the southeastern Brazilian continental margin (Fig. 01) is controlled by coastal hydrodynamic processes, which are strongly associated with the local wind-driven current system, by the northward-directed plume of the Río de La Plata on the inner and mid shelf, and by the meandering eddy structure of the southward flowing Brazil Current on the outer shelf (de Mahiques et al., 2004; 2008; 2011). Apart from the detection of the presence of sediments derived from the Río de La Plata as far north as 25°S, few studies have addressed material sources, as well as transport and sedimentation mechanisms in this region (de Mahiques et al., 2008; Razik et al., 2015). The understanding of sediment sources, transport and deposition is nevertheless of high relevance in this region, because major parts of the continental shelf have originally been considered sediment-starved (Kowsmann and Costa, 1979).

The origin and fate of regional continental slope surface sediments in the tropical and subtropical Atlantic were evaluated by Govin et al. (2012) by determining the concentrations of major elements (Ca, Fe, Al, Si, K, Ti) and some common metal/metal ratios. In their study, the values of ln(Ti/Al) and ln(Fe/K) were considered as valuable proxies for the identification of sediment sources, and an allochthonous character of the sediments on the continental slope off southeast Brazil was suggested. However, the study had limited sensitivity for the dynamics of regional circulation patterns.

Razik et al. (2015), using 25 sediment surface samples collected between 20°S and 45°S, proposed a conceptual source-to-sink sediment model for the SW Atlantic region, enhancing the role of Pampean (Argentina) originated sediments for the sediment cover on the SW Brazilian slope. It is, however, important to note that their water mass distribution model is not in agreement with the complex hydrodynamic situation along the SW Atlantic continental margin, especially on the SE Brazilian slope, as discussed later.

In this context, it is important to highlight that water masses (i.e., bodies of water with similar physical properties), considered as major players by the preceding studies, do not physically contribute to lateral transport of particles, unlike ocean currents (i.e., continuous and directed movement of a water mass). In particular, the South Atlantic Subtropical Gyre, during its movement toward the Brazilian continental margin, is influenced by both the wind field and the southern circulation in the South Atlantic. At the near-surface level (carrying Tropical Water–TW), the gyre is centred at 15°S. At the level of the thermocline (carrying South Atlantic Central Water–SACW), the gyre is centred at 20°S, following the main



lineament of the geomorphological Victoria–Trindade seamount chain. Finally, at the underlying intermediate level, between 500 and 1200 m water depth (carrying Antarctic Intermediate Water–AAIW), the gyre approaches the margin at 28°S (Schmid et al., 2000).

This slopeward-directed intermediate water flow leads to the occurrence of an unparalleled oceanographic phenomenon at

the upper slope, which is known as the Santos Bifurcation (Boebel et al., 1997; 1999). From a hydrodynamic point of view, the Santos Bifurcation leads to the formation of a permanent western boundary current (Intermediate Western Boundary Current–IWBC) (Boebel et al., 1999; Silveira et al., 2004; Legeais et al., 2013; Biló et al., 2014; Schattner et al., 2016) which leads to a reversed flow compared to the regional southward-directed circulation (Brazil Current–BC). Thus, a unique situation occurs in the global ocean occurs. In this situation, an intermediate water mass, transported by a boundary current,

runs opposite to the surface circulation, which leads to the generation of a transition zone in which the current velocities tend to be zero (Biló et al., 2014). According to Schmid et al. (2000), about one fourth of the 19 Sv that reach the Brazilian margin at the Santos Bifurcation turns to the north, thus diverting water from the South Atlantic Gyre. Another flow inversion occurs at 1200–1300 m water depth at the interface between the IWBC and Deep Western Boundary Current (DWBC), which transports North Atlantic Deep Water (NADW) toward the south (Stramma and England, 1999).

Considering the complex hydrodynamics of the SE Brazilian margin north of 28°S, this study analyses a new data set of both organic and inorganic sediment parameters, collected from the inner shelf to the middle slope. The following main questions are addressed:

- Despite the "sediment-starved" character off the inner shelf, is the final destination of coastal sediments determined by shelf dynamics in such a way that they are exported to the deep ocean?

- Can inorganic and organic geochemical signatures help to differentiate coastal from shelf and slope sediments?

- Is it possible to differentiate distinct geochemical signatures in the sediments that reflect the various ocean current systems on the SE Brazilian slope, namely the BC, IWBC and DWBC?

## 2 Study Area

### 2.1 Geology

The study area corresponds to the central part of the São Paulo Bight, as defined by Zembruscki (1979). It extends from 23°30'S to 26°00'S and from 44°00'W to 46°00'W (Fig. 1). A complex coastline, including Precambrian headlands, islands consisting of crystalline rocks, pocket beaches and small-scale bays and estuaries, defines the inner boundary of the study area. The latter occur in coastal plains and filled with Eemian (Marine Isotope Stage 5e) and mid- to late Holocene sea-level highstand sediments (Suguio and Martin, 1978).

The southeastern Brazilian margin has its origin in the geological evolution as an Atlantic-type continental margin. It is marked by subsidence of the Santos Basin (Cretaceous to recent) (Meisling et al., 2001) associated with intensive Mesozoic



alkaline magmatism, and by uplift of the Serra do Mar mountain range (Almeida and Carneiro, 1998). The maximum thickness of the sedimentary successions reaches 12 km, represented by a variety of depositional environments ranging from terrestrial to deep ocean (Nunes et al., 2004).

## 2.2 Oceanography

The southern Brazilian upper margin is characterized by the confluence of two main current systems. The outer shelf and upper slope are dominated by the southward flow of the Brazil Current (BC) (de Mahiques et al., 2004). The inner shelf, in contrast, is controlled by the northward-flowing Brazilian Coastal Current (BCC) (de Souza and Robinson, 2004). The BCC transports in part sediments from the Rio de la Plata (de Mahiques et al., 2008) river plume (Möller et al., 2008). This seasonal wind-dependent northward displacement of colder and less saline waters also controls the variability in regional

primary productivity of the area (Ciotti et al., 1995).

Late Holocene sedimentation rates on the continental shelf and slope vary from 2 to 68 cm ka$^{-1}$ and are controlled by local shelf and upper-slope morphology, meandering dynamics of the BC and offshore movement of the Coastal Water (CW) (de Mahiques et al., 2011). The latter is a result of upwelling dynamics, occasionally driven by passing BC meanders and local wind conditions. In the southern sector of the shelf, sedimentation is strongly influenced by terrigenous supply from the

remote Rio de la Plata outflow, which leads to the formation of a number of inner- and mid-shelf mud depocenters (de Mahiques et al., 2016). The lowest sedimentation rates occur on the outer shelf and around the shelf break, confirming the strong dependency of these sedimentary processes on the coupled BC–IWBC system (de Mahiques et al., 2011). The sediment distribution exhibits latitudinal variations and bathymetric control. The area off São Sebastião Island (24°S–45°30'W) marks a regional boundary between two main sedimentary provinces that are characterized by differences in

organic (bulk $\delta^{13}C$) and inorganic (clay mineralogy, metals) compositions (Nagai et al., 2014a; 2014b).

South of São Sebastião Island (24°S), depositional processes are related to the northernmost penetration of shelf waters associated with the Rio de la Plata freshwater plume. This coastal plume, carried northwards along the southern Brazilian shelf by the BCC (de Souza and Robinson, 2004), has been identified as far north as 24°S (Campos et al., 1999). However, Nd and Pb isotope data indicate that the influence of the Rio de la Plata outflow is strongly limited to 28°S, with the sector

between 28°S and 24°S being a transitional zone (de Mahiques et al., 2008).

North of 24°S, i.e. off São Sebastião Island, the heterogeneity of sediments is related to more complex hydrodynamics, especially on the inner and middle shelf, where a diversified ocean structure interacts with the local seafloor morphology, the presence of islands, the concave shape of the coastline and the presence of a BC branch meander, which leads to a highly complex oceanographic situation (Silveira et al., 2000; Dottori and Castro, 2009).

In addition, the sedimentary processes exhibit a conspicuous bathymetric control. For instance, on the inner shelf, Holocene sedimentation has levelled the irregular relief which itself developed during lower sea level (Nagai et al., 2014a), whilst on



the outer shelf, low sedimentation rates and very local sand fields are associated with the main flow of the BC which acts as a "floor-polisher" here (de Mahiques et al., 2004).

On the continental slope, the bathymetric configuration generally indicates a variable morphology, with slope gradients between 1:10 and 1:100. An amphitheatre-shaped feature is developed southeast of São Sebastião Island (24°30'S–44°00'W and 26°30'S–46°00'W), where alternating concave and convex isobaths contours indicate a history of a partly failing prograding slope process (Maia et al., 2010; Reis et al., 2013).

The coverage of sediment samples on the slope is still incipient and heavily concentrated on its upper part. The shelf break/upper slope transition zone shows sandy sediments with high concentrations of biolithoclastic and bioclastic sediments (de Mahiques et al., 2011). The sparse sediment samples from the middle and lower slope do not allow for a proper characterization of the sedimentary cover. Seismic lines interpreted by Maia et al. (2010) nevertheless show wavy and chaotic internal reflectors that indicate gravity-driven mass movement deposits in areas with a higher slope gradient.

## 3 Material and Methods

In this study, we used 107 sediment samples collected from eight multi-cores (MUCs) and one Van Veen grab sampler, from the inner shelf to the middle slope off the SE Brazilian margin (Table 1, Fig. 1). The MUC samples were sliced in 1-cm intervals, frozen and later freeze-dried.

Approximately 10 mg of planktonic foraminifera *Globigerinoides ruber* and *G. sacculifer* tests (>150-μm size fraction) from six of the cores (NAP-62, -63, -64, -65, -66, -68) were AMS dated at Beta Analytic Laboratories. Following Lund et al. (2015), we applied a reservoir age $\Delta R$ of $7 \pm 200$ years (1σ) to account for unknown changes in the reservoir age through time. Calendar ages were calibrated using Calib v.7.1 (http://calib.qub.ac.uk/calib/calib.html) (Table 2). Core-tops were considered as modern since it was possible to verify that the sediment-water interfaces remained intact during core sampling. Metals (Al, Ba, Ca, Fe, K, Mg, Sc, Sr, Ti, V) were analysed with a Varian 710 ICP-OES, following the procedures of total digestion established in the US Environmental Protection Agency 3052 Method. About 0.25 g of dry sediment was weighted and inserted in a Teflon digestion tube, together with 10 mL of concentrated $HNO_3$. The mixture was digested by microwave-digestion for 20 minutes. Then, 2 to 5 mL of 40% HF were added and the digestion procedure was repeated. Finally, 1 mL of 30% $H_2O_2$ was added and the digestion procedure was repeated once more. The solution was then filtered through Whatman no. 40 filters. The filtrate was transferred to a Teflon beaker and heated on a hot plate until evaporation. Then, 20 mL of 5% $HNO_3$ were added three times and the remnant residue was transferred to a 50-mL flask with distilled water and the solution analysed by ICP-OES. The accuracy of the results was verified by digestion of aliquots of the NIST Estuarine Sediment SRM1646a certified standard.

For data analysis and interpretation of metal concentrations, two approaches were used in this study. First, we used the same logarithm of the ratios presented in Govin et al. (2012), with the exception of the ln(Al/Si), since Si was not analysed in this



study. We also used the Ca/Al ratio (Sun et al., 2008) and K/Al ratio (Plewa et al., 2012), here presented in their logarithmic forms, as secondary approximations for the relative contribution of sediment sources.

A second approach consisted of a Principal Component Analysis (PCA), using metal contents. The PCA was performed on the correlation matrix, using PAST version 3.14 software (Hammer et al., 2001).

Bulk organic parameters ($C_{org}$ and $N_{tot}$ concentrations; $\delta^{13}C_{PDB}$ and $\delta^{15}N_{AIR}$ isotopes) were determined using a Finnigan Delta V Plus mass spectrometer coupled with a Costech Elemental Analyser. Reference materials used to control the quality of data obtained are presented as Supplementary Material (Supp. Mat 1). $CaCO_3$ content was determined by weight difference of the sediments before and after acidification with 10% HCl.

For the analysis of $n$-alkanes we used the analytical procedure employed by Bicego et al. (2006). Three g of dry sediment
were extracted with methylene chloride and $n$-hexane (1:1, v/v) using a Soxhlet apparatus, after adding $n$-hexadecene and $n$-eicosene as surrogates. Aliphatic hydrocarbons and alkenones were separated from the total lipid fraction by column chromatography using 5% deactivated alumina (1 g) and silica (2 g). The elution was done with 6 mL of $n$-hexane and 10 mL of DCM:$n$-hexane (9:2, v/v). Tetradecene was then added as an internal standard.

Aliphatic hydrocarbons were analysed by gas chromatography (Agilent GC model 6890) by flame ionization detection (GC-
15 FID). The column used was a fused silica column coated with 5% diphenyl/dimethylsiloxane (50 m, 0.32 mm ID and 0.17 mm film thickness). Hydrogen was used as the carrier gas. The oven temperature was programmed to increase from 40°C to 60°C at 20°C mi$n$-1, to 250°C at 5°C mi$n$-1, to 300°C at 20°C mi$n$-1 and finally to 320°C at 6°C mi$n$-1, where it was held for 20 min. Aliphatic hydrocarbons ($n$-alkanes from $C_{12}$ to $C_{34}$, phytane and pristane) were identified by matching the retention times with data measured using standard mixtures of HCs from AccuStandard (USA).

Quality assurance and quality control were performed based on the analysis of the procedural blank, blank spike, matrix spike, standard reference material (IAEA-417, International Atomic and Energy Agency) and internal standard recovery. The average recovery for internal standards was 72% (50–106%) and recovery in the spike blank and spiked samples varied from 62% to 101% and from 51% to 112%, respectively. Concentrations found in the blanks were subtracted from all samples. Certified standards at five different concentrations were injected to build analytical curves with correlations of at least 0.995.
The method detection limit (LDM) was based on the standard deviation of seven replicates and varied from 0.005 to 0.017 µg g-1.

The following $n$-alkane-based indexes were used in this study:

1. Terrigenous/Aquatic ratio (TAR) (Bourbonniere and Meyers, 1996):

$$TAR = \frac{\sum C_{27+29+31}}{\sum C_{15+17+19}}$$

This index is based on the assumption that photosynthetic algae and bacteria are characterised by short-chain $n$-alkanes ($C_{17}$) when compared with the $C_{27+29+31}$-rich molecules present in vascular plants. Bourbonniere and Meyers (1996) observed that despite the fact that the TAR index may over-represent the absolute amounts from terrigenous sources, it can be used as a qualitative proxy for the relative contributions of organic matter from both terrestrial areas and aquatic sources.



2. Terrestrial *n*-alkanes Index (TI) (this paper):

$$TI = \frac{\Sigma C_{27+29+31}}{C_{org}}$$

This estimation is based on the same assumption as described above (TAR), but is calculated as a ratio over the total organic

carbon ($C_{org}$) content. This parameter is based on the assumption that greater values of TI represent a higher input of

5    terrestrial organic matter.

3. Alkane Index (AI) (Schefuß et al., 2003):

$$AI = \frac{nC31}{nC31 + nC29}$$

Used here as a proxy for the climatic conditions in the source area for organic carbon, $C_{31}$ is assumed to be more common in

tropical dry areas than $C_{29}$, which is more abundant in temperate climates (Horikawa et al., 2010). According to Zhang et al.

10   (2006), AI values higher than 0.5 indicate an influence of grasses ($C_4$ plants) as organic carbon sources. On the other hand,

AI values lower than 0.5 indicate that the terrestrial organic matter is derived from $C_3$ plants (Rommerskirchen et al., 2006).

4. Average Chain Length (ACL) (Ficken et al., 2000):

$$ACL = \frac{\sum [Ci] * i}{\sum [Ci]}$$

Here, Ci is the *n*-alkane with i number of carbons, varying from 25 to 33 and describes the average number of carbon atoms

15   per molecule based on the abundance of the odd-carbo*n*-numbered higher plant *n*-alkanes (Poynter and Eglinton, 1990; Jeng,

2006). Used as a proxy of terrestrial organic matter input, values of ACL between 25 and 33 represent a higher contribution

of terrestrial plants (Collister et al., 1994; Rommerskirchen et al., 2006). The ACL values of *n*-alkanes can also be used to

show the distribution of $C_3$ and $C_4$ plants (Rommerskirchen et al., 2006; Vogts et al., 2009).

5. Carbon Preference Index (CPI) (Eglinton et al., 1962; Eglinton and Hamilton, 1963; Aboul-Kassim and Simoneit, 1996):

$$CPI = \frac{1}{2} \text{ x } \left( \frac{[n - C_{25}] + [n - C_{27}] + [n - C_{29}] + [n - C_{31}] + [n - C_{33}]}{[n - C_{24}] + [n - C_{26}] + [n - C_{28}] + [n - C_{30}] + [n - C_{32}]} \right.$$
$$\left. + \frac{[n - C_{25}] + [n - C_{27}] + [n - C_{29}] + [n - C_{31}] + [n - C_{33}]}{[n - C_{26}] + [n - C_{28}] + [n - C_{30}] + [n - C_{32}] + [n - C_{34}]} \right)$$

Even though this index is widely used to differentiate natural and anthropogenic sources of organic matter (Pietrogrande et

al., 2010), it is also considered a proxy of marine (CPI lower than 1.5) versus terrestrial (CPI higher than 2.3) contributions

(Collister et al., 1994; Bi et al., 2005).

25   6. Pristane/Phytane Ratio (Pri/Phy): in recent sediments, this parameter is usually used to identify the organic matter source

(Rashid, 1979). This ratio is also thought to be an indicator of depositional conditions and the relative degree of alteration or

degradation of organic matter (Powell and McKirdy, 1973; Didyk et al., 1978). Here it is used as a proxy of bottom redox

conditions. According to Didyk et al. (1978), values of Pr/Ph lower than one indicate anoxic water above anoxic sediments;

on the contrary, fully oxic conditions provide Pr/Ph values higher than one. As mentioned above, we also performed a PCA

30   on the data, using the molecular parameters cited above as variables.



## 4 Results and Discussion

All cores dated display late Holocene ages (Table 2). Assuming that the core-tops are modern as explained above, calculated average sedimentation rates vary from 3.4 cm kyr$^{-1}$ (core 64) to a maximum of 14.7 cm kyr$^{-1}$ (core 62) (Table 2). Worth to note, core 65 (taken at 121-m water depth), also showing a young age, is located in an area previously described as covered

by relict sediments (de Mahiques et al., 2004). Indeed, visual inspection of the core showed stained and oxidized bioclasts as well as limonite-covered quartz grains, indicative for relict sediment facies (Wilson, 2010).

Results of both inorganic and organic analyses are presented as Supplementary Materials 2, 3, and 4. Figures 2 to 11 present values as average and standard deviations from the full depths of each core.

### 4.1 Inorganic parameters

The ln(Ti/Ca) and ln(Ca/Al) ratios show the expected inverse patterns (Fig. 2). Lower values of ln(Ti/Ca and higher ln(Ca/Al) ratios (-4.44 ± 0.19 and 1.81 ± 0.16, respectively) were measured in core 65 (121 m water depth), which corresponds to the relict facies mentioned above. A higher terrigenous influence was found in core 61 (60 m water depth, ln(Ti/Ca) = -2.06 ± 0.06; ln(Ca/Al) = -0.51 ± 0.10), located on the inner shelf. In general, mid-slope sediments (samples 67 and 68) show a higher terrigenous contribution when compared with the outer-shelf and upper-slope samples.

The ln(Ti/Al) and ln(Fe/K) ratios (Fig. 3) show a higher variability, especially on the continental slope. The ln(Ti/Al) ratio allows a clear distinction of samples 67 (649 m, -2.81 ± 0.02) and 68 (1393 m, -2.80 ± 0.06) from the rest, including sample 63, which is located in an intermediate water depth (840 m, -2.66 ± 0.08). In the same way, samples 67 and 68 show distinct ln(Fe/K) ratios (0.88 ± 0.08 and 0.93 ± 0.03, respectively) compared to the rest of the samples, except for sample 65, indicating a stronger kaolinitic influence. Samples from the shelf and upper slope show higher K contents (lower values of

ln(Fe/K)).

Differing compositions of shelf and mid-slope sediments is also observed in the plot of ln(K/Al) versus ln(Fe/Ca), especially considering the former parameter. Once more, there is a discrepancy of samples 67 (649 m) and 68 (1393 m) when compared with sample 63, which is located in a bathymetrically intermediate position.

The two first PCA components for the metal data (Fig. 5) account for 89.8% of the total variance (1st: 79.0%; 2nd: 10.8%).

Results clearly separate the carbonate-rich relict sediments on the outer shelf (core 65, 121 m water depth) from the siliciclastic sediments from both shelf and slope. In addition, while the sediments on the shelf and upper slope are rich in K, Ba and Ti, sediments on the deeper slope are relatively enriched in V, Fe and Mg. It is important to note that the sediments on the middle slope (core 63, 840 m water depth) exhibit a transition between these two geochemical characteristics.

### 4.2 Organic Parameters



We used the N/C ratio as a bulk organic proxy indicating terrestrial input (Perdue and Koprivnjak, 2007; Biscara et al., 2011) and compared it with the $\delta^{13}$C (Fig. 6) and $\delta^{15}$N values (Fig. 7). In both figures, it is possible to recognize two distinct patterns, corresponding to shelf and upper slope samples on one side of the plots, and middle slope sediments on the other. Shelf and upper slope sediments show an expected increase of N/C ratios associated with increasing $\delta^{13}$C (Fig. 6) and $\delta^{15}$N

values (Fig. 7). On the other hand, middle slope samples show no variation in N/C ratios, independent of the isotope values. This distinct aspect is probably related to the depletion of nitrogen in the deeper sediments off Brazil (Metzler et al., 1997), as well as to the availability of nitrogen during seasonal upwelling processes on the shelf (Sumida et al., 2005; Moser et al., 2014).

With regard to molecular markers, TAR ratios (Fig. 8) are higher than one for all of the studied samples, indicating a

contribution of terrestrial higher plants to the study area, especially visible in core 61, located on the inner shelf. The TAR values range between 1.0 and almost 20.0 on the shelf, with the widest range occurring solely in the innermost shelf sediments, while the slope shows TAR signals typically between 1.0 and 4.8. A similar trend of TAR distribution patterns can be deduced from the terrestrial *n*-alkane Index (TI) (Fig. 8) with values higher than 100 µg gC$^{-1}$ found in the inner shelf samples. In addition, a rapid progressive TI decrease toward deeper water is detected with minimum values around 4.4 µg

gC$^{-1}$.

Average chain length (ACL) values (Fig. 9) in all samples range from 28.96 to 33.00, showing the presence of terrigenous influence in all sites. This ratio can also be used to infer the vegetation types that are dominant in the terrigenous source. Leaf lipids derived from grasslands have longer chain lengths than leaf lipids from forest plants (Cranwell, 1973). In rainforest C$_3$ plants, *n*-C$_{29}$ dominates, whereas the dominant *n*-alkanes in C$_4$ grasses are *n*-C$_{31}$ and *n*-C$_{33}$ (Rommerskirchen et

al., 2006; Badewien et al., 2015). The average ACL values for sample 61 in shelf waters are typical of C$_3$ plant influence and lower than those from the slope, mainly characterized by *n*-alkanes derived from C$_4$ plants, which is corroborated by the AI ratio.

The Alkane Index (AI) (Fig. 9) also shows a clear differentiation between shelf and slope, with the former showing an AI range between 0.25 and 0.56 and the latter with values higher than 0.55 (with the singular exception of core 63 at 840 m

water depth). The AI has been used to investigate dominant inputs from C$_4$ and C$_3$ terrestrial plants. Vegetation dominated by grasses and herbs (C$_4$ plants) normally presents AI values larger than 0.5. On the other hand, forest assemblages show AI values less than 0.5 (Schefuß et al., 2003). Accordingly, the organic matter of cores 61 and 62, located on the shelf is dominated by input of C$_3$ vegetation derived from the Atlantic rainforest adjacent to the sampling site. The cores from the slope show AI index values typical for grasses, which indicates that there is a difference in the contribution from terrigenous

sources between slope and shelf, thus defining the limits of the contribution of the adjacent continent to sediment organic matter.

The Carbon Preference Index (CPI) (Fig. 10) is tool that indicates the *n*-alkane source. Normally, CPI values above three are associated with a strong predominance of compounds from land plant waxes (Clark Jr and Blumer, 1967; Hedges and Prahl, 1993), while values close to one indicate *n*-alkanes mostly derived from phytoplankton, bacteria, recycled organic matter or



petroleum (Clark Jr and Blumer, 1967; Volkman et al., 1983; Kennicutt et al., 1987). From the CPI obtained for individual cores (Fig. 10) it is possible to observe that most samples from cores 61 and 62 show CPI values above, indicating a strong influence of terrigenous material. The other cores are characterised by lower CPI values, suggesting the contribution of either marine or recycled organic matter.

As a general rule, the Pristane/Phytane ratio (Pri/Phy) tends to increase with increasing contribution of terrestrial organic matter (Rashid, 1979). Figure 10 shows a trend of decreasing values toward deeper areas, with highest values in NAP 61, corresponding to both TAR and TI distributions. According to Didyk et al. (1978) values of Pr/Ph lower than one indicate anoxic water above anoxic sediments, while fully oxic conditions provide Pr/Ph values greater than one. This ratio can also be affected by the presence of bacterial material (Volkman and Maxwell, 1986; Ten Haven et al., 1988). The values obtained

in this study show a wide range in shelf waters, ranging between 1.3 and 2.8, while with increasing water depth, the values vary between 0.9 and 1.3, indicating a possible contribution of marine or recycled organic matter.

The two first components of the PCA analysis for the molecular parameters (Fig. 11) account for 75.6% of the total variance of the system and allow to synthetize the variations of terrestrial versus marine or recycled organic matter as well as of the relative contribution of C3 and C4 plants. Component 1 (58.0% of the variance) distinguishes between shelf and slope

samples, the first being characterized by higher values of TAR, TI, CPI and Pri/Phy, as well as lower values of ACL and AI. On the other hand, Component 2 (17.6% of the variance), separates samples from the upper and middle slope, the latter marked by the highest values of ACL.

**4.3 Sediment sources and ocean circulation**

The southeastern Brazilian shelf has been frequently classified as a "starving platform", with modern sediment cover

considered the result of reworking transgressive sediments. Nevertheless, it is worth noting that there are mud depocenters, as defined by Hanebuth et al. (2015), in which muddy sedimentation can be significant and Holocene sedimentation rates can be higher than 50 cm kyr$^{-1}$ (de Mahiques et al., 2011; Mendoza et al., 2014; de Mahiques et al., 2016). The sources of these sediments are still a matter of debate, but a mixture of local and distant sources cannot be discarded (Mendoza et al., 2014).

The ln(Ti/Al) versus ln(Fe/K) ratios (Fig. 3), ln(K/Al) versus ln(Fe/Ca) ratios (Fig. 4), and the PCA of the metals (Fig. 5)

indicate separate geochemical signatures for the shelf and slope sediments in the study area. The outer shelf relict facies is marked by higher Ca and Sr concentrations as well as a ln(Fe/K) ratio higher than 0.8. The coastal and regional shelf and upper slope signatures correspond to higher contents in Ti and K. In this sense, and considering the humid climatic conditions along the southeastern coast of South America, the coastal and shelf sediment values might reflect proximity to the source rock, in contrast to previous interpretation attempts linking element provenance patterns to limited chemical

weathering on the adjacent continent (Mulitza et al., 2008; Just et al., 2012; Nace et al., 2014; Razik et al., 2015). Important to note is the fact that the geochemical signature of the shelf sediments is similar to that of the upper slope, at least down to the 370-m isobath, hinting to widespread continental supply even without a major fluvial source.





A more complex scenario is found for the sediments on the middle and lower continental slope. Considering all the inorganic proxies analysed, the sediments of core 63 (840 m water depth) are geochemically distinct from those of cores 67 (649 m water depth) and 68 (1393 m water depth), which themselves are geochemically very similar to each other. The execution of a no*n*-parametric Kruskal–Wallis ANOVA analysis also documents the significant differences between core 63 and the two others. According to Schmid et al. (2000), core 63 is in the area of the Intermediate Western Boundary Current (IWBC), which is located between 800 m and 1100 m water depth carrying Antarctic Intermediate Water (AAIW) to the north at these latitudes. This flow is in opposition to both the southward flowing Brazil Current (BC), which carries South Atlantic Central Water (SACW), as well as the Deep Western Boundary Current (DWBC), which transports North Atlantic Deep Water (NADW). Thus, the individual geochemical signal found at this particular location seems to be related to the nucleus of this intermediate current, delivering material from a remote sediment source along a clearly defined isobathic pathway.

A comparison of our data with those presented in Govin et al. (2012) clearly shows that our data are comparable with their findings for the domains of the BC and DWBC, but essentially different from those from the intermediate levels of the Argentinean margin, which, in fact, may extend to Uruguay (Franco-Fraguas et al., 2016). This leads us to propose that the divergent flows generated by the Santos Bifurcation (Boebel et al., 1999) lead to a geochemical differentiation, at intermediate levels, along the southwest Atlantic margin.

In this sense, the picture that emerges from our analysis is different from that presented by Razik et al. (2015). A possible explanation for this discrepancy might be related to the fact that the mechanisms thought to be responsible for lateral sediment transport in the present study are found in the complex vertical SW Atlantic current circulation, and not in its water mass structure as stated by those authors. In addition, the very limited number of samples (n = 2) used by Razik et al. for the same area might be one of the causes for this difference.

Finally, it is worth noting that samples from core 63 (840 m – IWBC) also slightly differ in the characteristics of the organic matter when compared with cores 67 (649 m – BC) and 68 (1393 m – DWBC) (Fig. 11), corroborating the observation of a distinct remote sediment (and organic matter) source under the domain of the IWBC.

### 4.4 Origin and characteristics of the organic matter

We tried to evaluate the relative contribution of terrestrial organic matter using the bulk organic $F_{ter}$ Index (Terrigenous Fraction) (Biscara et al., 2011), and we observe that our results are partially acceptable. With regard to N/C, the distinction between shelf/upper slope versus middle slope samples, including very high N/C values found in the shelf sediments may be an indicator of partial enrichment of both total and organic nitrogen during diagenetic processes (Müller, 1977; Álvarez-Salgado et al., 1997), as well as regeneration of nitrogen (Metzler et al., 1997). Concerning $\delta^{13}C$, Fter estimates vary between 70% on the shelf, to about 30% on the deeper slope, suggesting that the SW Atlantic margin can receive excessive amounts of terrestrial organic matter, despite the lack of major adjacent rivers.




In addition, molecular proxies for the origin of the organic matter similarly indicate a relatively high input of terrigenous organic matter to the ocean. Regional rainfall is higher than 2,000 mm yr$^{-1}$ (Fujieda et al., 1997), and the great potential of numerous small mountain rivers to supply terrestrial organic matter was already described by Leithold et al. (2006) and Goñi et al. (2013), among others, and was also addressed by de Mahiques et al. (1999) for the study area.

In order to compare the quality of the organic matter deposited on the area, plots of both ACL versus AI (Fig. 9) and CPI versus Pri/Phy (Fig. 10) evidence the difference between shelf and slope samples, with $C_3$ plants prevailing as the main source for terrigenous organic matter in the shelf sediments as indicated in previous studies (Rommerskirchen et al., 2006; Zhang et al., 2006). In contrast, almost all of the samples collected on the slope show AI values higher than 0.55, indicating that $C_4$ plants are the main source for the terrigenous fraction in the deeper deposits and, thus, suggesting that the source of

organic matter is distinct from that of the adjacent continent. The origin of this organic matter is still a matter of debate, since both BC and DWBC can transport sediments from further north.

## 5 Conclusions

In this study, we used inorganic and organic parameters in sediment cores from the southeastern South American margin to analyse both ocean current circulation and terrestrial material contribution patterns. The results reveal a regional input of

terrigenous sediments that covers both the continental shelf and upper slope, down to the 400-m isobath.

We were also able to differentiate the geochemical signatures associated with northward transport by the Intermediate Western Boundary Current, in comparison with the southward displacements of the Brazil Current and Deep Western Boundary Current.

Molecular markers indicate that input of terrestrial material is an important contributor to the organic matter deposited on

both shelf and slope. In addition, molecular proxies indicate important differences in the sources of terrestrial organic matter. Shelf and upper slope sediments show a higher contribution of $C_3$ plants, while $C_4$ plant material characterises the deeper slope sediments.

## 25  Acknowledgements

Financial support was provided by the São Paulo Science Foundation (FAPESP), grants 2010/06147-5 and 2015/21834-2, and the University of São Paulo via the NAP-GEOSEDEX Program. MMM acknowledges CNPq (grant 303132/2014-0). Thanks are due to the crew and researchers on board RV Alpha Crucis.



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





| Table 1. Position of the samples analyzed in this work | | | | | | |
|---|---|---|---|---|---|---|
| **Station** | **Sampling Device** | **Date (d/m/y)** | **LAT** | **LONG** | **Water Depth (m)** | **Sediment Depth (cm)** |
| 60 | Van Veen | 24/02/2013 | 23°32.745 S | 45°05.900 W | 16 | Surface |
| 61 | Multiple corer | 25/02/2013 | 23°57.600 S | 45°08.500 W | 52 | 23 |
| 62 | Multiple corer | 25/02/2013 | 24°02.532 S | 44°32.540 W | 121 | 28 |
| 63 | Multiple corer | 26/02/2013 | 24°50.400 S | 44°19.110 W | 840 | 35 |
| 64 | Multiple corer | 26/02/2013 | 24°37.891 S | 44°29.635 W | 302 | 15 |
| 65 | Multiple corer | 26/02/2013 | 25°11.363 S | 45°15.217 W | 121 | 21 |
| 66 | Multiple corer | 27/02/2013 | 25°36.291 S | 45°06.235 W | 368 | 15 |
| 67 | Multiple corer | 27/02/2013 | 25°38.844 S | 45°04.629 W | 649 | 40 |
| 68 | Multiple corer | 27/02/2013 | 25°47.532 S | 45°01.343 W | 1393 | 40 |

| Table 2. AMS radiocarbon datings of the multiple cores | | | | |
|---|---|---|---|---|
| **Sample** | **Lab ID** | **Conventional $^{14}$C age (yr BP) ± 1$\sigma$** | **Median probability and 2$\sigma$ cal range (yr BP)** | **Average Sedimentation rates (cm/kyr)** |
| NAP62-2_22 | 368261 | 1880 +/- 30 | 1445 (1250 - 1487) | 17.1 |
| NAP63-2_25 | 368262 | 4330 +/- 30 | 4450 (4183 - 4508) | 6.0 |
| NAP64-2_16 | 368263 | 4560 +/- 30 | 4750 (4513 - 4812) | 3.6 |
| NAP65-2_19 | 368264 | 2420 +/- 30 | 2060 (1824 - 2106) | 10.2 |
| NAP66-2_23 | 368265 | 5140 +/- 30 | 5475 (5296 - 5551) | 4.4 |
| NAP68-2_37 | 368266 | 3230 +/- 30 | 3045 (2780 - 3095) | 12.6 |





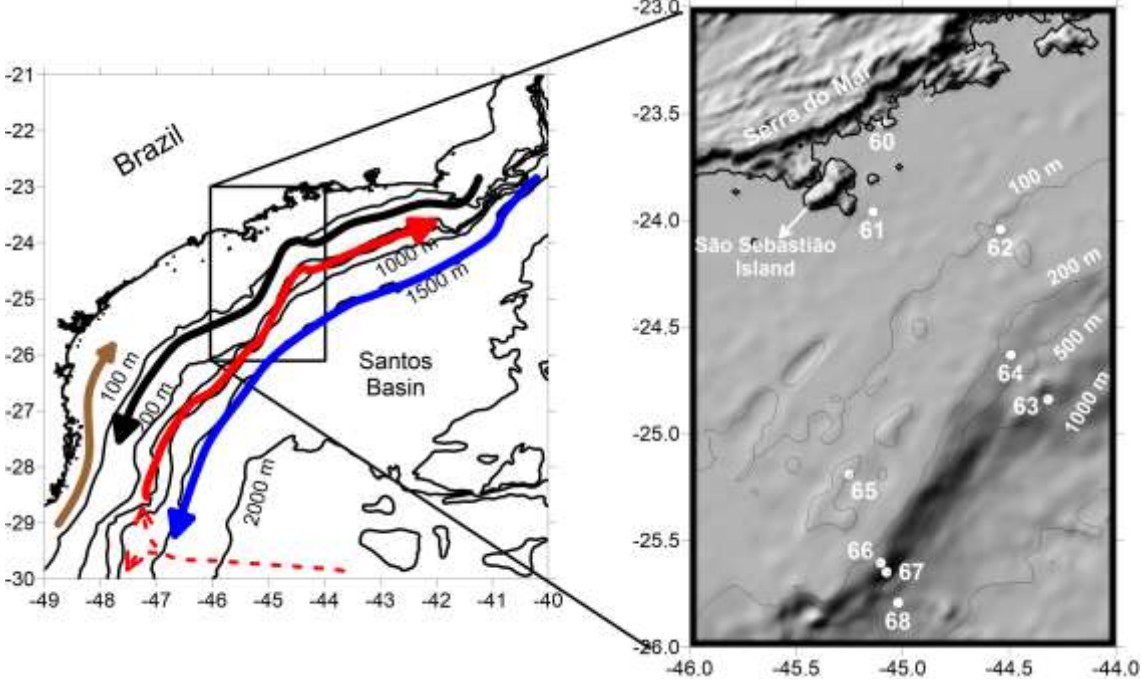

Fig. 1. Location of the study area and sampling stations. Arrows refer to the main current systems in the area: Brazilian Coastal Current (BCC) - brown; Brazil Current (BC) - black; Intermediate Western Boundary Current (IWBC) - red; Deep Western Boundary Current (DWBC) - blue. Dashed red line corresponds to the approximate position of the Santos Bifurcation (Boebel et al., 1999).





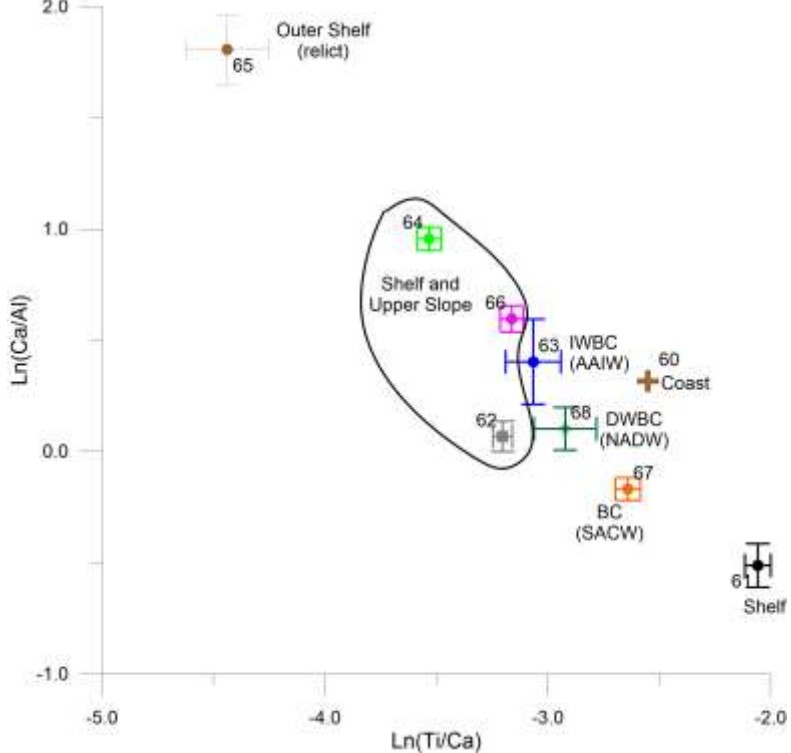

Figure 2. Scatter plot of average ± standard deviation of Ln(Ca/Al) versus Ln(Ti/Ca) values. Abbreviations: BC: Brazil Current; IWBC: Intermediate Western Boundary Current; DWBC: Deep Western Boundary Current; SACW: South Atlantic Central Water; AAIW: Antarctic Intermediate Water; NADW: North Atlantic Deep Water





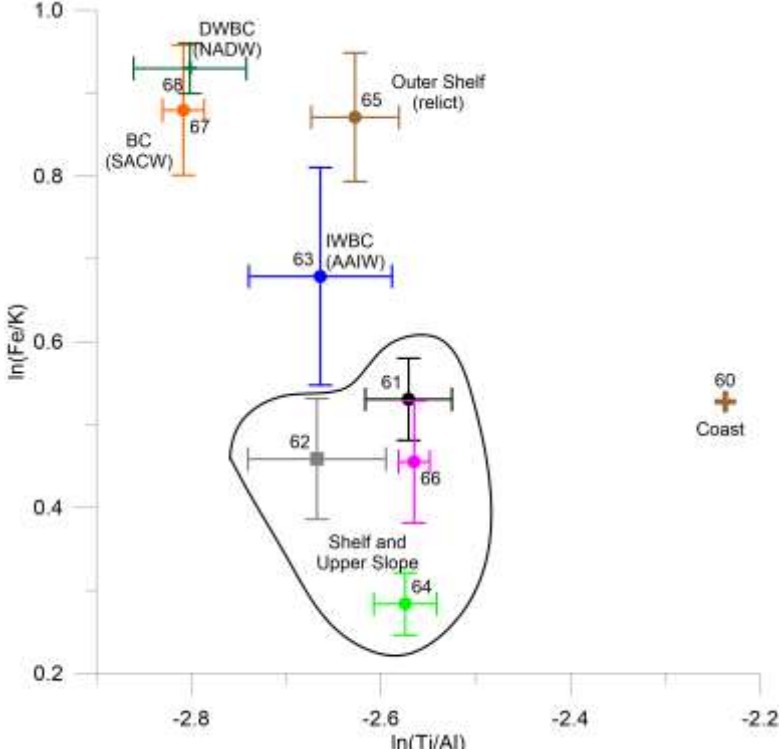

Figure 3. Scatter plot of average ± standard deviation of Ln(Fe/K) versus Ln(Ti/Al) values. Abbreviations: BC: Brazil Current; IWBC: Intermediate Western Boundary Current; DWBC: Deep Western Boundary Current; SACW: South Atlantic Central Water; AAIW: Antarctic Intermediate Water; NADW: North Atlantic Deep Water




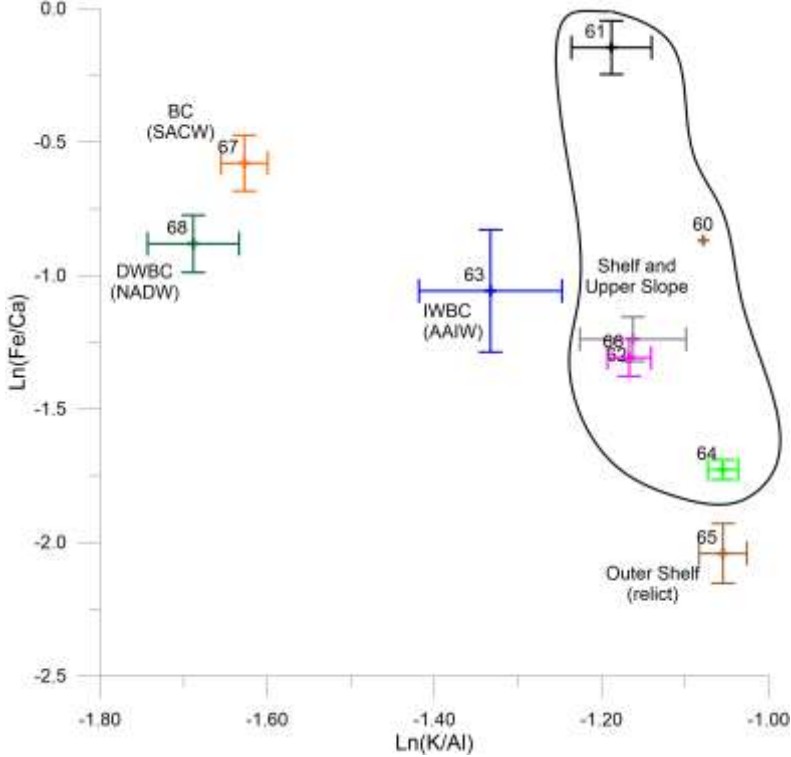

Figure 4. Scatter plot of average ± standard deviation of Ln(Fe/Ca) versus Ln(K/Al) values. Abbreviations: BC: Brazil Current; IWBC: Intermediate Western Boundary Current; DWBC: Deep Western Boundary Current; SACW: South Atlantic Central Water; AAIW: Antarctic Intermediate Water; NADW: North Atlantic Deep Water



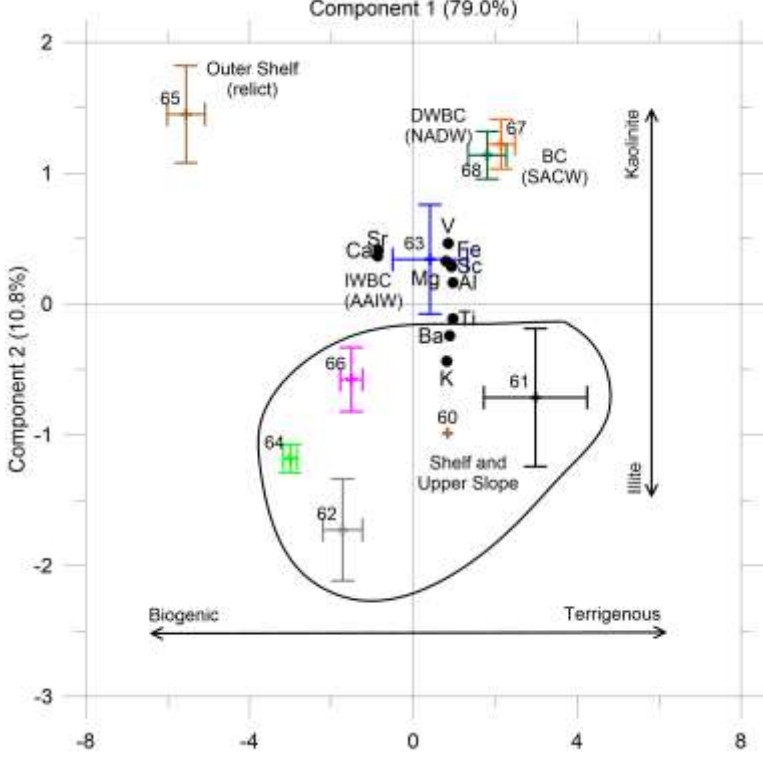

Figure 5. Biplot (samples and variables) of average ± standard deviation of the 1st and 2nd Components of the Principal Component Analysis of metals. Abbreviations: BC: Brazil Current; IWBC: Intermediate Western Boundary Current; DWBC: Deep Western Boundary Current; SACW: South Atlantic Central Water; AAIW: Antarctic Intermediate Water; NADW: North Atlantic Deep Water





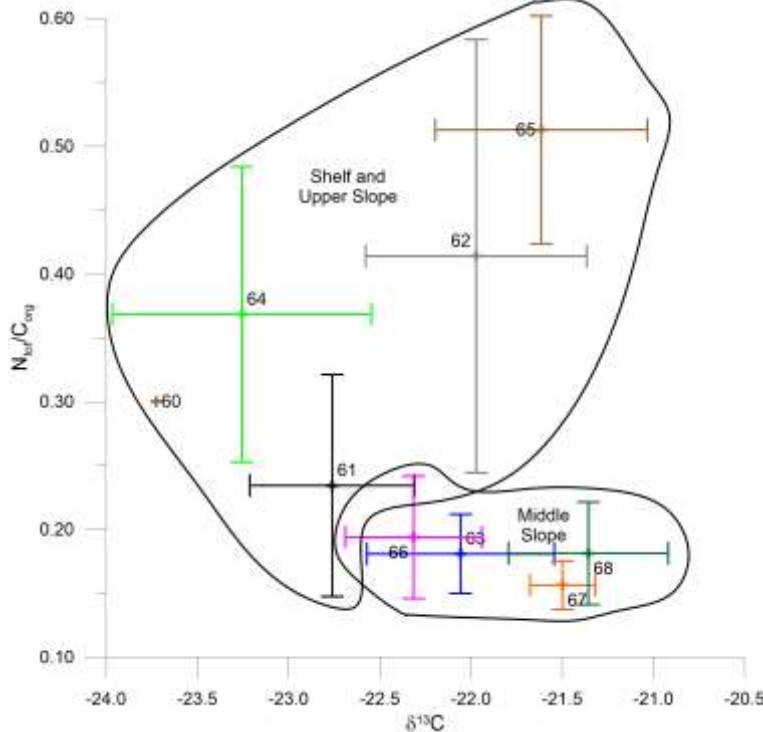

Figure 6. Scatter plot of average ± standard deviation of $N_{tot}/C_{org}$ versus $\delta^{13}C$ values. Abbreviations: BC: Brazil Current; IWBC: Intermediate Western Boundary Current; DWBC: Deep Western Boundary Current; SACW: South Atlantic Central Water; AAIW: Antarctic Intermediate Water; NADW: North Atlantic Deep Water



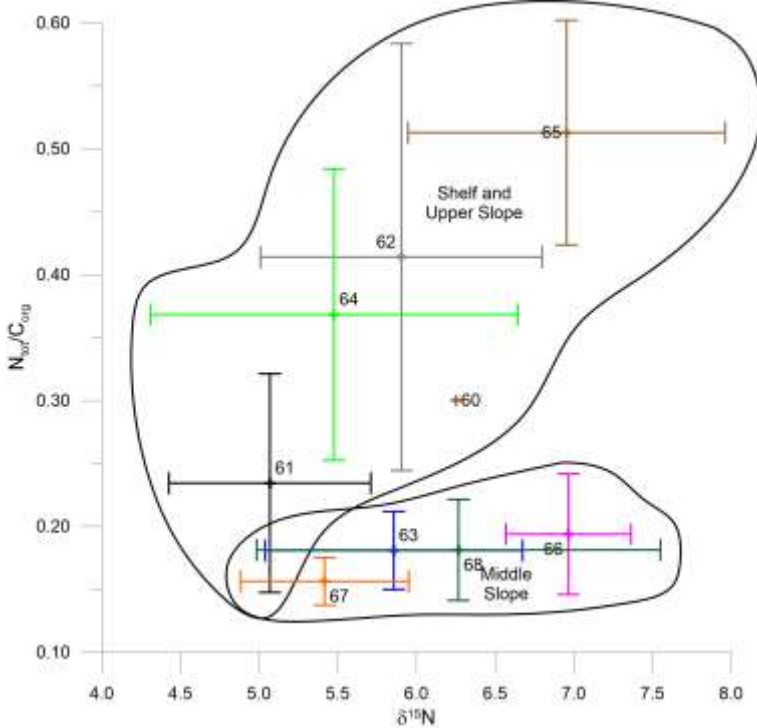

Figure 7. Scatter plot of average ± standard deviation of $N_{tot}/C_{org}$ versus $\delta^{15}N$ values. Abbreviations: BC: Brazil Current; IWBC: Intermediate Western Boundary Current; DWBC: Deep Western Boundary Current; SACW: South Atlantic Central Water; AAIW: Antarctic Intermediate Water; NADW: North Atlantic Deep Water



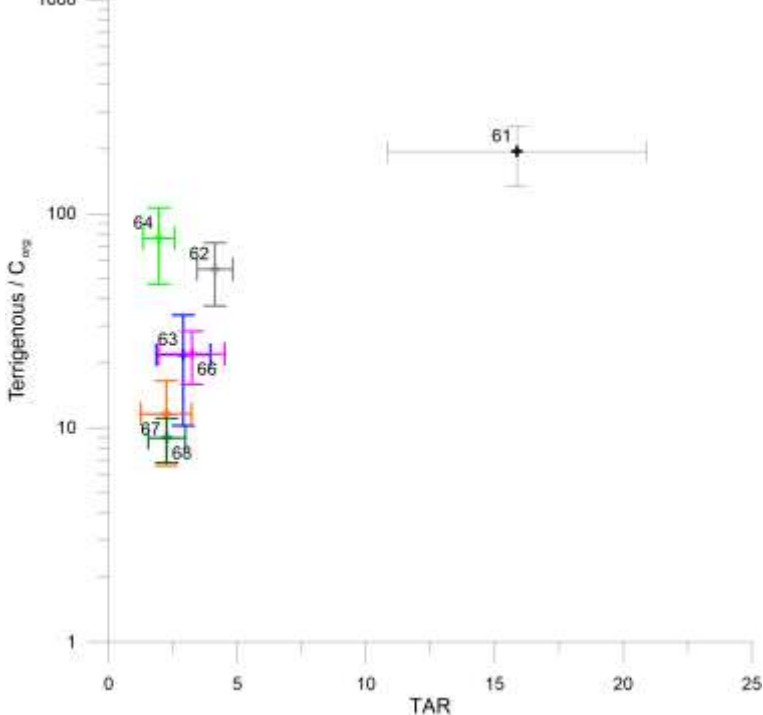

Figure 8. Scatter plot of average ± standard deviation of TI versus TAR values. Abbreviations: BC: Brazil Current; IWBC: Intermediate Western Boundary Current; DWBC: Deep Western Boundary Current; SACW: South Atlantic Central Water; AAIW: Antarctic Intermediate Water; NADW: North Atlantic Deep Water




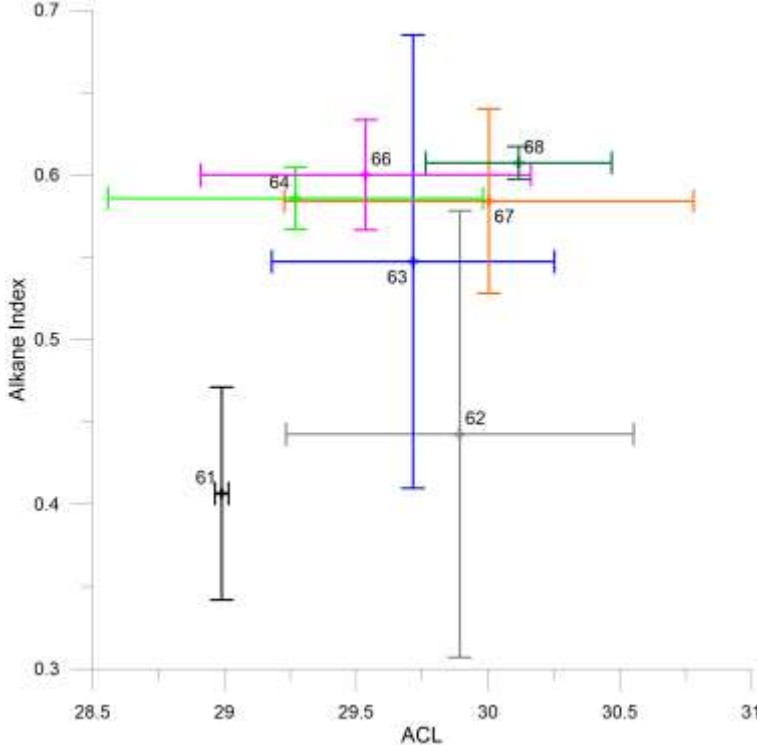

Figure 9. Scatter plot of average ± standard deviation of AI versus ACL values. Abbreviations: BC: Brazil Current; IWBC: Intermediate Western Boundary Current; DWBC: Deep Western Boundary Current; SACW: South Atlantic Central Water; AAIW: Antarctic Intermediate Water; NADW: North Atlantic Deep Water





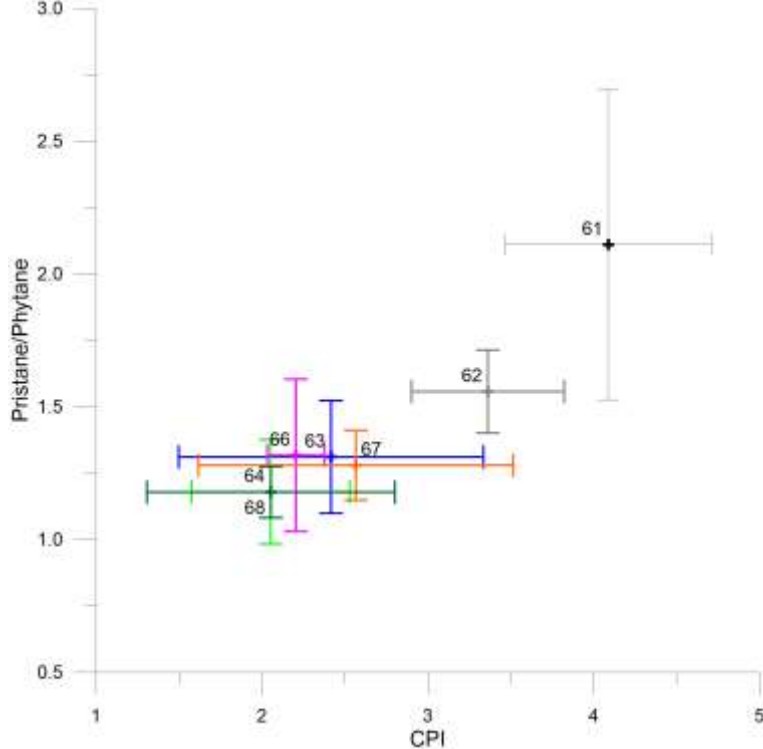

Figure 10. Scatter plot of average ± standard deviation of Pri/Phy versus CPI values. Abbreviations: BC: Brazil Current; IWBC: Intermediate Western Boundary Current; DWBC: Deep Western Boundary Current; SACW: South Atlantic Central Water; AAIW: Antarctic Intermediate Water; NADW: North Atlantic Deep Water





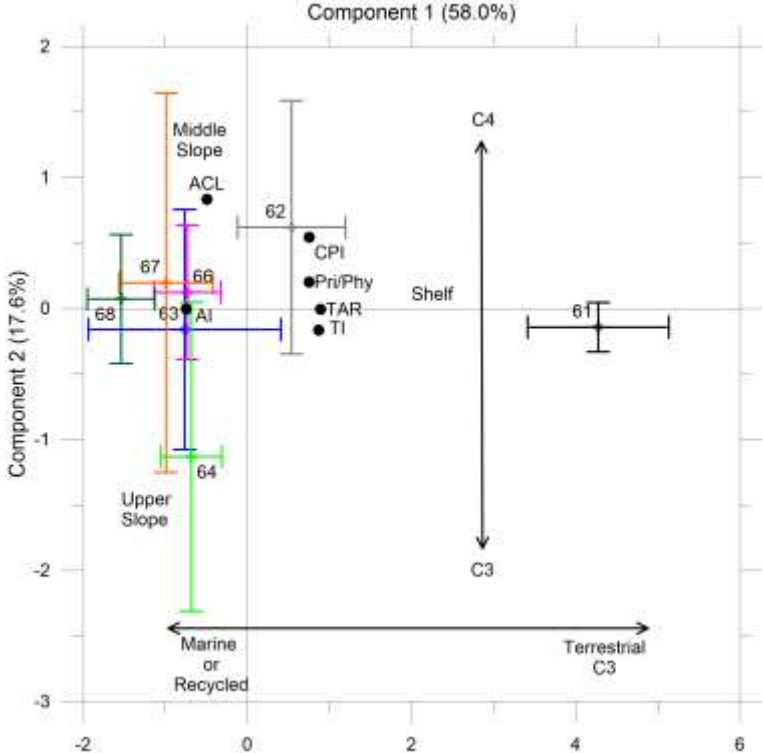

Figure 11. Biplot (samples and variables) of average ± standard deviation of the 1st and 2nd Components of the Principal Component Analysis of molecular markers. Abbreviations: BC: Brazil Current; IWBC: Intermediate Western Boundary Current; DWBC: Deep Western Boundary Current; SACW: South Atlantic Central Water; AAIW: Antarctic Intermediate Water; NADW: North Atlantic Deep Water