# Peer review of "Inorganic and organic geochemical fingerprinting of sediment sources and ocean circulation on a complex continental margin (São Paulo Bight, Brazil)"

_Ocean Science, 2016_

## Referee Comment (RC1) · N. Fagel (Referee) · 21 Dec 2016

General comment - The topic of the ms fits well with a publication in Ocean Sciences. Both the problematic and the methodology is well described and complete. I would suggest some changes, mainly on the figures, in order to better support the interpretations. In addition, some part of the discussion could be better explain. Taking into account my comments, I would suggest a moderate revision. See below the point-by-point review.

Point-by-point review
- Lines 27-29 page 2 and Fig. 1 - Introduction– I would suggest to present the vertical distribution of the water masses along the latitudes. It is difficult with the two figures to relate each coring site with a water mass.
- Line 8 page 2 – repetition of "occurs" to remove.
- Lines 18-19 page 3 - unclear sentence that must be rewritten (*"in such a way that they are exported to the deep ocean"*).
- Line 31 page 3 – the text into bracket "Cretaceous to Recent" to place before subsidence.
- Line 18 page 5 – Are you shure about the reservoir age "we applied a reservoir age of 7+/-200yrs"? I suggest to add the reservoir age in the table 2. May be useful to get the distribution of probability for the calendar ages, rather than just the interval.
- Figures 2-5 – The caption must be expanded. The relationship between each coring site and a specific current and/or water mass must be presented in a figure (see previous comment on figure 1). The figures 2, 3 and 4 may be merged or present in parallel.
- Figure 5 - The caption must be expanded to resume the significance of the relative position of the cores by respect with the position of the inorganic elements. How do you explain that Ca and Sr remain both quite close to zero along the x axis whereas some cores are plotted at-3 (64) or even -6 (65)?
- Figures 6 & 7 – By comparison with N/C, the $\delta15N$ is a less discriminant parameter. I would suggest to plot all data with different symbols on the same diagram as the y axis is the same.
- Figure 8 – Is this diagram really useful? The information on average TAR values may be given in the text.
- Figure 9 – same remark. I'm not shure this diagram is useful. If you keep it you should add the domains for C3 and C4 plant and labelled the cores according to their location (upper shelf/slope). For all figures, you could systematically use a sme symbol for the cores according to their location on the upper shelf, middle shelf and slope. It could help to better evidence the data clusters.
- Lines 2-5 page 8 – the section about relict sediment must be rewritten, it is not clear.
- Lines 3-9 page 10 – This section is unclear. What about the oxic/anoxic conditions of the bottom water column according to the measured Pr/Ph values? To better explain.
- Lines 14-18 page 11 - This text is confuse. You may develop your argumentation.
- Final remark – I would suggest to add at the end of the ms a synthetic figure to present the sedimentary transport and their relationships with current distribution.

Hoping this review will be useful,
Sincerely,
Nathalie Fagel

---

## Referee Comment (RC2) · Anonymous Referee #2 · 23 Jan 2017

This manuscript focused on the source and chemistry of sediment delivered to a portion of the continental shelf and slope off the eastern coast of Brazil. The goal was to determine the provenance of sediment in an area of the coastal ocean impacted by complicated ocean currents. Inorganic and organic parameters were measured, including a range of trace metal concentrations, organic carbon and nitrogen content, stable carbon and nitrogen isotopes, and long-chain n-alkanes and pristane/phytane ratios.

The manuscript is well-written and conveys the main points of the results well. These

data will likely be interesting to a range of readers of Ocean Sciences and I advocate for its eventual publication here. However, I do have some suggestions that the authors should address before the paper is published. I've organized those suggestions here by paper section.

Introduction In general, I found the introduction to be unsatisfactory. Except for the first paragraph, the section was simply an overview of the study site, replicated almost completely in Section 2 (Study Area). The authors should remove all of the summary of the oceanographic setting (all of the Introduction except for Paragraph 1) and ensure that all of that information is in Section 2.2. Then, there needs to be an introduction of the use of the parameters employed here in similar oceanographic studies with expanded information about the specific tracers, inorganic and organic, that are used here and the types of information that they can convey.

Study Area The study area description was very good, including both the geology and oceanography of the site. However, I would like Figure 1 (study area map) to be updated and improved to better match with these descriptions. Overall, I think including a larger area on Figure 1 would be useful, with things like Rio de la Plata, the South Atlantic Subtropical Gyre, and other currents that are mentioned in the Introduction section included. The inset with the study site descriptions was very good, but if these currents and water sources are important to interpreting the study data, it would be nice to see them laid out on the figure.

Materials and Methods Because many of the parameters used in this study were not described in the Introduction, the authors describe the use of these proxies in the Methods section. I'd like to see the descriptions of the purposes and use of these proxies moved to the Introduction section (where they can be expanded upon and described in greater detail) and just have laboratory and analytical methods included in this section.

Terrestrial n-alkanes Index: I am not sure of the utility of this index. Why include it when

you are already including previously used indices like the TAR and ACL? The TAR has the benefit of accounting for relative changes in total n-alkane input and degradation, while the newly introduced TI does not. Why introduce it here? If it is used, the authors should clearly describe its benefits for this study area that the many other indices they employed do not have.

Results and Discussion In general, I thought this section included many more figures and data than was appropriate. I am more familiar with the organic geochemical parameters than the inorganic, but many of the parameters seemed redundant. I think in general the authors should pick only a few parameters to show on scatter plots, even if they still use all of the data for the PCA analyses.

Core 65: In the beginning of the Results section, this core is described as having a "young age," which according to the radiocarbon data it does. However, for the rest of the manuscript it is described as containing mostly relict sediment. While this is what previous studies have shown, and what some of your inorganic parameters seem to support, the organic data and radiocarbon dates do not support this. I would like to see more of an explanation for this discrepancy in the data. It is possible that the mineral portion of the sediment has a different source than the organic (sorbing / desorbing processes, etc) but this should be explored by the authors in the Discussion.

ACL values (Fig 9): In the text describing this figure, it is mentioned that all samples range from 28.96 to 33.00, but in Figure 9, samples range to only 31.00 (33 would be off the plot axis).

AI description: the authors describe this as a proxy for C3 vs C4 plants, which is true. But, there is little attempt to use this information to help explain carbon source, aside from saying that the likely C3 source is the adjacent continent. Within the context of what is known about the ocean circulation here, where would the C4 signature be coming from? What is the vegetation like in the watersheds of nearby rivers? Explain where the C4 signature is coming from.

Section 4.3, first paragraph: I think there should be more explanation for this sentence: "...considering the humid climatic conditions along the southeastern coast of South America, the coastal and shelf sediment values might reflect proximity to the source rock, in contrast to previous interpretation attempts linking element provenance patterns to limited chemical weathering on the adjacent continent." Why do the authors make this assertion?

Section 4.3, third paragraph: The authors say, "In this sense, the picture that emerges from our analysis is different from that presented by Razik et al (2015)." How so? Please summarize the findings of that study and explain exactly how your findings are different in the text of your discussion?

Section 4.4, first sentence: "... we observe that our results are partially acceptable." How was this evaluated? What exactly do you mean by "partially acceptable"? Please be specific and explain exactly how this was evaluated

---

## Author Comment (AC1) · 1 Mar 2017

Dear Dr Chapman, Thank you for the opportunity to submit a revised version of our paper entitled Inorganic and organic geochemical fingerprinting of sediment sources and ocean circulation on a complex continental margin (São Paulo Bight, Brazil) First of all, we would like to thank to Dr Fagel and the anonymous reviewer for the revision and suggestions. Please find below the point-by-point answers and/or comments:

Referee 1:

- Lines 27-29 page 2 and Fig. 1 - Introduction– I would suggest to present the vertical

distribution of the water masses along the latitudes. It is difficult with the two figures to relate each coring site with a water mass.

We changed Figure 1 in order to include two bathymetric profiles, in which the vertical position of the samples and water masses is shown. We hope this is in accordance with the reviewer requirement

- Line 8 page 2 – repetition of "occurs" to remove.

Removed from the text

- Lines 18-19 page 3 - unclear sentence that must be rewritten ("in such a way that they are exported to the deep ocean").

The sentence has been changed

- Line 31 page 3 – the text into bracket "Cretaceous to Recent" to place before subsidence.

Done, as requested

- Line 18 page 5 – Are you shure about the reservoir age "we applied a reservoir age of 7+/-200yrs"? I suggest to add the reservoir age in the table 2. May be useful to get the distribution of probability for the calendar ages, rather than just the interval.

The reviewer is correct. Indeed, it is not the reservoir age but the regional correction (△R) to the global reservoir. The text has been changed. Table 2 presents the Median Probability and the 2ïĄş (Confidence Interval) for each dating. This is an acceptable output from CALIB (http://calib.org)

- Figures 2-5 – The caption must be expanded. The relationship between each coring site and a specific current and/or water mass must be presented in a figure (see previous comment on figure 1). The figures 2, 3 and 4 may be merged or present in parallel.

Figures have been put in parallel and renumbered in the manuscript. We also grouped the samples according to their location, such as "coast and inner shelf", "outer shelf", etc.

- Figure 5 - The caption must be expanded to resume the significance of the relative position of the cores by respect with the position of the inorganic elements. How do you explain that Ca and Sr remain both quite close to zero along the x axis whereas some cores are plotted at-3 (64) or even -6 (65)? We grouped the samples as "coast and inner shelf", "outer shelf" etc. We hope this is satisfactory

- Figures 6 & 7 – By comparison with N/C, the $\delta$15N is a less discriminant parameter. I would suggest to plot all data with different symbols on the same diagram as the y axis is the same.

We tried to attend this request but the resulting figure became very confuse. In this sense, we opted to present these scatter plots in a single file as Figures 4A and 4B

- Figure 8 – Is this diagram really useful? The information on average TAR values may be given in the text.

Figures have been put in parallel. We understand that the maintenance of the graphics is important for the comprehension of the manuscript

- Figure 9 – same remark. I'm not shure this diagram is useful. If you keep it you should add the domains for C3 and C4 plant and labelled the cores according to their location (upper shelf/slope). For all figures, you could systematically use a same symbol for the cores according to their location on the upper shelf, middle shelf and slope. It could help to better evidence the data clusters.

We adopted the suggestion of the reviewer and grouped the cores according to their location. Concerning the Alkane Index, our data show that there is a change of the source of organic matter to deeper areas, from C3 to C4 plants, the last coming from east and northeast Brazil (references added in the text)

- Lines 2-5 page 8 – the section about relict sediment must be rewritten, it is not clear.

We rewrote the text and hope it is clear now

- Lines 3-9 page 10 – This section is unclear. What about the oxic/anoxic conditions of the bottom water column according to the measured Pr/Ph values? To better explain.

Actually, there are no indications of an anoxic environment but the variations of the Pr/Ph point to an increase of recycling of the organic matter towards deeper areas. We have changed the text and added references

- Lines 14-18 page 11 - This text is confuse. You may develop your argumentation.

The text has been changed, in order to clarify our arguments

- Final remark – I would suggest to add at the end of the ms a synthetic figure to present the sedimentary transport and their relationships with current distribution.

We accepted the suggestion and inserted Figure 7, which is a conceptual model for the input of both organic and inorganic materials to the study area

Referee 2 This manuscript focused on the source and chemistry of sediment delivered to a portion of the continental shelf and slope off the eastern coast of Brazil. The goal was to determine the provenance of sediment in an area of the coastal ocean impacted by complicated ocean currents. Inorganic and organic parameters were measured, including a range of trace metal concentrations, organic carbon and nitrogen content, stable carbon and nitrogen isotopes, and long-chain n-alkanes and pristane/phytane ratios. The manuscript is well-written and conveys the main points of the results well. These data will likely be interesting to a range of readers of Ocean Sciences and I advocate for its eventual publication here. However, I do have some suggestions that the authors should address before the paper is published. I've organized those suggestions here by paper section. Introduction In general, I found the introduction to be unsatisfactory. Except for the first paragraph, the section was simply an overview of the study site, replicated almost completely in Section 2 (Study Area).

- The authors should remove all of the summary of the oceanographic setting (all of the Introduction except for Paragraph 1) and ensure that all of that information is in Section 2.2. Then, there needs to be an introduction of the use of the parameters employed here in similar oceanographic studies with expanded information about the specific tracers, inorganic and organic, that are used here and the types of information that they can convey.

A significant part of the Introduction has been moved to the Study area and we added two paragraphs on the utilisation of proxies in the recognition of source areas and sediment transport. Nevertheless, it is important to highlight that the focus of the paper is on the SW Atlantic and its currents system as carriers of sediments. In this sense, we kept the information that was crucial for our objectives

- Study Area The study area description was very good, including both the geology and oceanography of the site. However, I would like Figure 1 (study area map) to be updated and improved to better match with these descriptions. Overall, I think including a larger area on Figure 1 would be useful, with things like Rio de la Plata, the South Atlantic Subtropical Gyre, and other currents that are mentioned in the Introduction section included. The inset with the study site descriptions was very good, but if these currents and water sources are important to interpreting the study data, it would be nice to see them laid out on the figure.

We changed Figure 1 in order to attend to both referees

- Materials and Methods Because many of the parameters used in this study were not described in the Introduction, the authors describe the use of these proxies in theMethods section. I'd like to see the descriptions of the purposes and use of theseproxies moved to the Introduction section (where they can be expanded upon and described in greater detail) and just have laboratory and analytical methods included in this section.

We've added some information to the introduction, but if we'd move all of the requested text to the introduction the result would be an unbalanced text, when comparing with

the inorganic fraction. We are kindly asking to the reviewer to accept our arguments

-Terrestrial n-alkanes Index: I am not sure of the utility of this index. Why include it when already including previously used indices like the TAR and ACL? The TAR has the benefit of accounting for relative changes in total n-alkane input and degradation, while the newly introduced TI does not. Why introduce it here? If it is used, the authors should clearly describe its benefits for this study area that the many other indices they employed do not have.

As stated in other parts of this reply, these figures have been put in parallel. Concerning the TI index, as seen in Figure 5A, it helped to discriminate the samples from the different compartments. Just as an example, this index shows that the amount of C27-29-31 (per gC) in the inner shelf is about 16 times higher than on the middle slope, which gives an idea of the dilution of terrestrial organic matter in deeper areas.

- Results and Discussion In general, I thought this section included many more figures and data than was appropriate. I am more familiar with the organic geochemical parameters than the inorganic, but many of the parameters seemed redundant. I think in general the authors should pick only a few parameters to show on scatter plots, even if they still use all of the data for the PCA analyses.

Despite the opinion of both reviewers, the authors understand that the graphic information is easier to be assimilated by the reader. In this sense, instead of simply removing figures, we decided to put several in parallel and, thus, keeping the information available. Only the figures of the PCA were kept in their original forms. We ask the reviewers to consider our decision as satisfactory.

- Core 65: In the beginning of the Results section, this core is described as having a "young age," which according to the radiocarbon data it does. However, for the rest of the manuscript it is described as containing mostly relict sediment. While this is what previous studies have shown, and what some of your inorganic parameters seem to support, the organic data and radiocarbon dates do not support this. I would like to see

more of an explanation for this discrepancy in the data. It is possible that the mineral portion of the sediment has a different source than the organic (sorbing / desorbing processes, etc) but this should be explored by the authors in the Discussion.

Sample 65 is essentially a carbonate sand (please take a look in our Ti/Ca ratio for this sample). We couldn't even analyse it for n-alkanes because the amount of organic carbon was too low. Also, it presents limonite-stained grains and shells. This area was already reported as a relict facies in one of our previous studies (de Mahiques et al., 2004) and this relict character is maintained by the "floor-polisher" effect of the Brazil Current. In this sense, we changed the text, in order to explain better our arguments

- ACL values (Fig 9): In the text describing this figure, it is mentioned that all samples range from 28.96 to 33.00, but in Figure 9, samples range to only 31.00 (33 would be off the plot axis).

All of the figures present the mean and $\pm$ 1 sigma. That's why the ranges are different. The value of 33.00 cited is beyond the 1sigma limit

- AI description: the authors describe this as a proxy for C3 vs C4 plants, which is true. But, there is little attempt to use this information to help explain carbon source, aside from saying that the likely C3 source is the adjacent continent. Within the contexto of what is known about the ocean circulation here, where would the C4 signature be coming from? What is the vegetation like in the watersheds of nearby rivers? Explain where the C4 signature is coming from.

We included information about this subject and added corresponding references

-Section 4.3, first paragraph: I think there should be more explanation for this sentence: ". . .considering the humid climatic conditions along the southeastern coast of South America, the coastal and shelf sediment values might reflect proximity to the source rock, in contrast to previous interpretation attempts linking element provenance patterns to limited chemical weathering on the adjacent continent." Why do the authors

make this assertion?

This sentence has been changed in order to clarify

-Section 4.3, third paragraph: The authors say, "In this sense, the picture that emerges from our analysis is different from that presented by Razik et al (2015)." How so? Please summarize the findings of that study and explain exactly how your findings are different in the text of your discussion?

The sentence was changed in order to better clarify

- Section 4.4, first sentence: ". . . we observe that our results are partially acceptable." How was this evaluated? What exactly do you mean by "partially acceptable"? Please be specific and explain exactly how this was evaluated

This was removed from the text. It was an attempt to compare the contribution of terrestrial organic carbon based on the N/C ratio and the $\delta$13C. But previous studies already point to problems in the utilization of the N/C (or C/N) ratios to estimate the % of terrestrial contribution in the area.

Once more, we thank both reviewers and we hope that the new text is in an acceptable form to be published in Ocean Sciences.

Michel M de Mahiques On behalf of the authors

[Figure]

[Figure]

**Fig. 2.** Figure 2. Scatter plots of average ± standard deviation of A) Ln(Ca/Al) versus Ln(Ti/Ca); B) Ln(Fe/K) versus Ln(Ti/Al); C) Ln(Fe/Ca) versus Ln(K/Al). Colours refer to samples from: coast and inner she

[Figure]

[Figure]

**Fig. 4.** Figure 4. Scatter plots of average ± standard deviation of A) Ntot/Corg versus $\delta 13C‰$ and B) $\delta 15C‰$ versus $\delta 13C‰$. Colours refer to samples from: coast and inner shelf (brown), outer shelf and u

[Figure]

**Fig. 5.** Figure 5. Scatter plot of average ± standard deviation of A) TI versus TAR, B) AI versus ACL, C) Pr/Ph versus CPI. Colours refer to samples from: coast and inner shelf (brown), outer shelf and upper s

[Figure]

[Figure]

---

## Author Response (AR2)

Dear Dr Chapman

Thank you once more for your careful read of four manuscript.

We are submitting the version that included all of the changes requested.

Kind Regards

Michel M de Mahiques

On behalf of the authors

---

## Author Response (AR3)

Dear Dr Chapman

We moved the paragraph, as requested.

Kind Regards

Michel M de Mahiques

On behalf of the authors